# Biomimetic and Materials-Potentiated Cell Engineering for Cancer Immunotherapy

**DOI:** 10.3390/pharmaceutics14040734

**Published:** 2022-03-29

**Authors:** Tingting Zhang, Yushan Yang, Li Huang, Ying Liu, Gaowei Chong, Weimin Yin, Haiqing Dong, Yan Li, Yongyong Li

**Affiliations:** 1Shanghai Skin Disease Hospital, School of Medicine, Tongji University, Shanghai 200092, China; z1778233513@tongji.edu.cn (T.Z.); 2031092@tongji.edu.cn (Y.Y.); 2131252@tongji.edu.cn (L.H.); 1931236@tongji.edu.cn (Y.L.); 1931235@tongji.edu.cn (G.C.); yinweimin@tongji.edu.cn (W.Y.); yongyong_li@tongji.edu.cn (Y.L.); 2Key Laboratory of Spine and Spinal Cord Injury Repair and Regeneration, Ministry of Education, Tongji Hospital, School of Medicine, Tongji University, Shanghai 200092, China

**Keywords:** immune cell, biomimetic engineering, materials-potentiated cell engineering, cancer immunotherapy

## Abstract

In cancer immunotherapy, immune cells are the main force for tumor eradication. However, they appear to be dysfunctional due to the taming of the tumor immunosuppressive microenvironment. Recently, many materials-engineered strategies are proposed to enhance the anti-tumor effect of immune cells. These strategies either utilize biomimetic materials, as building blocks to construct inanimate entities whose functions are similar to natural living cells, or engineer immune cells with functional materials, to potentiate their anti-tumor effects. In this review, we will summarize these advanced strategies in different cell types, as well as discussing the prospects of this field.

## 1. Introduction

Immune cells, such as T cells, natural killer (NK) cells, neutrophils, macrophages and dendritic cells (DCs), are the main functional components in cancer immunotherapy. They can recognize and directly kill tumor cells (e.g., T cells, NK cells, neutrophils and macrophages) or present antigens to effector T cells to trigger the anti-tumor immune response (e.g., DCs). However, owing to the taming of the tumor immunosuppressive microenvironment, the anti-tumor functions of these immune cells are significantly inhibited. A series of immune checkpoints (e.g., PD-1 [1], adenosine [2]), metabolism dysregulation in the tumor microenvironment (TME) (e.g., amino acid metabolism [3], lipid metabolism [4]), immunosuppressive cells (e.g., regulatory T cells (Tregs) [5], tumor-associated macrophages (TAMs) [6]) and immunosuppressive cytokines (e.g., TGF-β, IL-6, IL-10), could cause the dysfunction of these immune cells by exhaustion, or phenotype differentiation.

A variety of strategies have been reported to maintain the anti-tumor phenotype and enhance the anti-tumor immune response of immune cells; for example, systemic injection of cytokines or antibodies [7,8], or the utilization of genetic engineering technology to reform immune cells before reinfusion, such as Chimeric Antigen Receptor T-Cell (CAR-T cell). However, due to the toxicity caused by non-specific distribution and the safety issues of genetic engineering technology [9], it is urgent to develop new strategies to promote cancer immunotherapy.

Materials, especially nanomaterials have made remarkable progress in tumor immunotherapy. They can be constructed from various man-made organic or inorganic materials, or natural cell components, such as the cell membrane or cell-secreted vesicles. Due to the fantastic functions of nanomaterials, the biostability, bioavailability and biosafety of various small molecular drugs, cytokines, ligands, aptamers and antibodies could largely be improved. Some even realize intelligent and controllable therapeutics release on demand. Moreover, by elaborate design, materials with special functional groups can be used as a bridge to integrate the functional molecules with immune cells to endow immune cells with tailored or new functions. In recent years, various materials-engineered strategies have been reported for cancer immunotherapy (Figure 1). These strategies mainly focused on two parts: biomimetic strategies and materials-potentiated cell engineering strategies. Biomimetic strategies utilize bioactive cell-derived components or biomimetic materials, as building blocks to construct inanimate entities whose functions are similar to natural living cells (Table 1). The abilities of targeting [10,11], chemotaxis [12], invisibility [13], antigen presentation [14], phagocytosis and killing can be achieved via cell-mimicking strategies [15,16]. Moreover, the biomimetic entities can be used as a “decoy”, to attract the immunosuppression signals [15,17,18]. Therefore, the immunosuppression of immune cells in TME could be relieved and immune activation could be indirectly achieved.

Different from biomimetic strategies, the strategies of materials-potentiated cell engineering are to endow immune cells with different supportive components (also known as backpack strategies), or functional materials to enhance their anti-tumor effects, prevent and rescue their dysfunction, maintain and restore their anti-tumor phenotype (Table 2) [29,30,31]. The functional materials can be either coupled on the cell surface covalently/non-covalently or internalized into cells by physical extrusion, natural phagocytosis, and others. As the research on cells and immunity progresses, the fields of nanomaterials, medical immunity, and cell biology have largely become integrated, in terms of interdisciplinarity, which provides a new platform for cancer treatment. In this review, we will present the overview in the way of classification of cell types. Firstly, their anti-tumor mechanisms and challenges in immunotherapy are introduced. Then, we summarize the biomimetic and materials-potentiated cell engineering strategies for cancer immunotherapy. Finally, the prospects and development directions of these strategies will be prospected, to provide insights for future related study.

## 2. Biomimetic and Materials-Potentiated Cell Engineering Strategies

### 2.1. T Cell

T cells, especially cytotoxic T lymphocytes (CTLs), are the “main force” of anti-tumor immune cells. Upon activation by major histocompatibility complex (MHC) molecule-tumor antigen complex, CTLs can eliminate tumor cells, through the Fas-ligand-mediated apoptosis or the secretion of cytotoxic molecules (e.g., granzyme B) [58]. However, in tumor patients, especially in solid tumors, immunosuppressive TME hinders T cell infiltration and leads to functional exhaustion [59]. Therefore, various materials-engineered strategies were designed to mimic or enhance the functions of T cells. 

#### 2.1.1. T Cell Biomimetic Strategy

The Fas–FasL pathway is a central route for T cells to modulate tumor apoptosis. Nanoparticles containing camptothecin (CPT) and the anti-Fas antibody have been designed as T cell mimics by a sandwich structure (PEG as outer layer, anti-Fas antibody as middle layer, and a CPT-loaded inner core) (Figure 1) [20]. Synergized with CPT, the FasL-mediated cell apoptosis of the T cell mimics was up to 51.99%, which was significantly improved, compared to the control group without any treatment (almost no cell apoptosis).

In addition to directly killing tumor cells, T cell mimics could be fabricated as decoys to ameliorate immunosuppression and prevent excessive immune responses. Taking advantage of biomimetic materials, multiple immunosuppressive signals can be blocked. Eventually, the immune tolerance of the TME is disrupted and the infiltration of immune cells increases. Mukundan et al. reported a T cell mimic by constructing PD-1-conjugated microparticles (PDMPs) [21]. Unlike the antibody blockade, PDMPs could competitively inhibit the PD-1/PD-L1 signaling, while maintaining normal immune tolerance and the CD45^+^ immune cell recruitment was enhanced with PDMPs treatment, which was a mechanism for tumor control. 

Apart from that, integrating multiple components into one system could provide a “one-for-all” modality for combined immunotherapy. Zhang et al. encapsulated a 1-methyl-tryptophan (1-MT), an indoleamine-2,3-dioxygenase (IDO) inhibitor into engineered cellular nanovesicles, presenting PD-1 receptors (PD-1 NVs), to block the inhibitory effects of PD-1/PD-L1 and IDO on effector T cells, simultaneously. The percentage of activated CD8^+^ T cells in tumors from the PD-1 NVs-treated group was increased by 23.4%, compared to the PBS-treated group, and it further increased by 16.7% after the integration of 1-MT, which directly drove tumor regression. Besides, they genetically engineered platelets to express PD-1 and loaded a low dose of cyclophosphamide (CP) [17]. Such engineered platelets could masquerade as T cells, binding to PD-L1 on tumor cells via PD-1. Therefore, the PD-1/PD-L1 pathway between T cells and tumor cells is inhibited and, thus, the exhausting of T cells is inhibited. Accumulating in the surgery wound, the PD-1-expressing platelets and their derived microparticles increased the emergence of CD8^+^Ki67^+^GzmB^+^ lymphocytes and selectively depleted Tregs. With all these synergistic effects, the postsurgical tumor recurrence and metastasis were reduced.

Due to the presence of cell-derived components, natural cell membranes or cell-derived nanovesicles have multiple anti-tumor mechanisms. T-cell-derived nanovesicles (TCNVs) with programmed receptors could be produced by the continuous micro/nano pore extrusion of cells. TCNVs could block PD-L1 and clear TGF-β, and directly killed tumor cells by delivering granzyme B [19]. Similarly, Kang et al. developed T-cell-membrane-coated nanoparticles (TCMNPs), which exhibited higher therapeutic efficacy than PD-L1 antibodies in melanoma treatment, due to the multiple therapeutic mechanisms and the tumor-targeting ability (Figure 2a) [16]. By mimicking the mechanism of CTLs, TCMNPs released anti-tumor molecules, induced Fas-ligand-mediated apoptosis and removed immunosuppressive signals by the exhibited membrane molecules (Figure 2b). Besides, as a drug-carrying platform, TCMNPs could also be combined with chemotherapeutic drugs or a CTLA-4 blockade, to exert synergistic therapeutic effects.

#### 2.1.2. T Cell Engineering Strategy

Several T cell engineering therapies, including tumor-infiltrating lymphocyte (TIL) therapy, T cell receptor (TCR)-T cell therapy and CAR-T cell therapy, have achieved great success. However, the challenges of adoptively transferred T-cell therapy remain, in terms of the declined vitality and functions of T cells in solid tumor treatment [60]. Many chaperone molecules (e.g., supportive cytokines [7,31,32], TME immunomodulators) are used to maximize the efficiency and persistence of adoptively transferred cells in vivo. They also suffer from the risks of off-target toxicity, caused by the systemic administration of the high and sustained systemic levels that are needed for desired efficacy. Another approach is to secrete these supporting factors or knockout the inhibitory molecules genetically [61,62], which is also hindered by the cost barriers and uncontrolled risks.

Utilizing materials-potentiated cell engineering strategies, nanoparticles containing chaperone molecules could bind to the surface of T cells and target tumor sites simultaneously. This allows the delivery of different types and predetermined doses of chaperone molecules (including small molecules that cannot be recombined and expressed by cells), while minimizing the toxicity risk caused by the uncontrolled stimulation of transferred cells.

One of the most common strategies is to couple cytokine-containing nanoparticles to adoptively transferred T cells, to maintain their phenotype, function and lifespan. The multilamellar lipid nanoparticles encapsulating IL-15 super-agonist (IL-15Sa) and IL-21 were bound to the surface of T cells, through thiol-reactive maleimide headgroups [31]. The in vivo proliferation and long-term persistence of nanoparticle-bound T cells greatly outperformed unmodified Pmel-1 T cells. Li et al. used anti-CD45, as specific anchors for the attachment of TCR-signaling-responsive nanogels to T cells [32]. The increased reduction potential in the T cell surface after antigen recognition led to nanogel collapse and IL-15Sa release. The expansion of IL-15Sa-backpacked T cells in tumors was 16-fold greater than T cells supported by systemic cytokine injections, and 1000-fold greater than T cells without cytokine support. In addition, they also extended this method to the delivery of IL-2 [33]. All these cytokine nanoparticle-conjugated T cells have achieved good tumor therapy effects and the supportive effect of cytokine nanoparticles is antigen independent [31], so the amplified therapeutic effect can be directly transferred to T cells.

In addition to cytokines, delivering TME immunomodulators to block negative regulatory signals is also effective in preventing or rescuing the emergence of hypofunctional CAR-T cells. However, delivering these drugs to immune cells within the TME represents a challenge. Adenosine, a metabolite of the TME, has been reported to suppress T-cell proliferation and IFN-γ secretion, by binding the A2a adenosine receptor (A2aR) on the activated T cells [61,63]. Based on this, researchers chemically conjugated CAR-T cell with multilamellar liposomal vesicles (cMLV), loaded with SCH-58261 (an A2aR-specific small molecule antagonist), named as CAR-T. cMLV (SCH), to prevent CAR-T hypofunction and reverse the hypofunctional tumor-residing T cells (Figure 3a) [30]. Relative to CAR-T + cMLV groups, the amount of colocalization of cMLV (SCH) and CAR-T cells (78.57% ± 26.7%) inside tumors significantly increased (Figure 3b). Subsequent analysis on day 2 after treatment showed that CAR-T. cMLV (SCH) had the highest T-cell engraftment (52.96% ± 15.5%) (Figure 3c). Moreover, higher intracellular IFN-γ expression was observed compared to other groups (Figure 3d). 

To boost anti-tumor immunity, molecular interaction regulation in the T-cell synapse is an important therapeutic strategy [64,65,66]. Despite some antibodies targeting cell surface receptors in immunological synapses, and small molecule compounds targeting intracellular signaling pathways at the proximal end of TCRs [67,68,69], their clinical success hinges on delivering therapeutic drug doses to the immunological synapse. Shp1 and Shp2 are key phosphatases, which could downregulate T-cell receptor activation in the synapse. For promoting TCR signaling, Stephan et al. covalently coupled maleimide-functionalized nanoparticles, loaded with NSC-87877 (a dual inhibitor of Shp1 and Shp2) to the T-cell membrane [34]. Interestingly, CD8^+^ T-cells surface-linked nanoparticles were dynamically redistributed from the uropod of migrating cells into the nascent immunological synapse during antigen recognition. Shp1/2 inhibitor-loaded nanoparticles enabled the efficient delivery of compounds into the T-cell synapse (5.7-fold higher levels than free drug); there was a 5.2-fold reduction in tumor burden compared to the untreated group, 10 days after the T-cell transfer.

Previous studies have demonstrated that T cell exhaustion has a lot to do with the suppressive metabolic state (e.g., hypoxia, nutrient deprivation) of TME, which likely compromises the T cell therapeutic outcome of the solid tumor [69,70]. Despite various metabolic intervention strategies based on metabolic modulators (e.g., T cells precondition in vitro [71,72], nanoparticles-based drug delivery [73]), the poor pharmacokinetic limitations of the drug and the immunosuppressive TME make them unsatisfactory effects [6,74]. Through elaborate design, an effective cholesterol-based metabolic intervention strategy was proposed by Zhang and coworkers. The function of T cells depends on the amount of cholesterol on the cell membrane for clustering TCRs and forming immunological synapses [75,76,77]. To induce rapid and sustained T cell activation, liposomal avasimibe (an inhibitor of the cholesterol-esterification enzyme acetyl-CoA acetyltransferase) was clicked onto the T cell surface by lipid insertion (named T-Tre/BCN-Lipo-Ava cells) to elevate the plasma concentration of cholesterol for TCR clustering [35]. Retaining avasimibe on the T cell surface provided avasimibe with autocrine- and paracrine-like mechanisms of action, for maximizing the antitumor effector functions of both transferred and endogenous T cells. Unlike unmodified T cells, T-Tre/BCN-Lipo-Ava cells showed larger TCR microclusters and the TCRs featured with a more compact nature in the center of the immunological synapse. Further, the synapse of T-Tre/BCN-Lipo-Ava cells could mature faster because of smaller average area than unmodified T cells. Glioblastoma complete eradication was achieved in three of the five mice that received surface anchor-engineered CAR-T cells.

Apart from enhancing the viability, the abilities of tumor homing and recognition of transferred T cells are also important. Unlike CAR-T and TCR-T, CTLs isolated from mice bearing melanomas, could be engineered with an IL-4 receptor-binding peptide (IL4RPep-1) through a non-genetically engineered method [36]. Simply, a dioleoylphosphatidylethanolamine (DOPE)-biological anchor was used as a membrane phospholipid-based linker for engineering. IL-4R is commonly up-regulated in melanoma and IL-4R-targeted CTLs showed higher binding to melanoma cells. With the higher levels of secreted granzyme B and IFN-γ, the IL-4R-targeted CTLs exerted a more rapid and robust effector response. 

The complex immunosuppressive microenvironment provides a wide range of intervention targets for improving the immunotherapy effect. Through elaborate design, various strategies, based on materials-engineered T cells, can be used to increase the amount of T cells and immunomodulatory drugs in the tumor site to achieve combined therapy. For clinical translation, safety and vitality maintenance of cell therapy products are very important. In addition to the continuous optimization of drug manufacturing solutions, researchers, surprisingly, found that frozen nanogel-loaded CAR-T cells retained the ability for cytokine-driven expansion, after thawing, which may be inspiring for drug storage [32].

### 2.2. NK Cell

NK cells, a unique subset of cytotoxic lymphocytes, are considered to be the most cytotoxic cells against tumors in vivo [78]. They have multiple mechanisms of effector function, including perforin/granzyme particles, antibody-dependent cell-mediated cytotoxicity, and cell death, mediated by death receptors and cytokines. In addition, the antitumor innate and adaptive immune responses can be further activated by proinflammatory cytokines and chemokines, secreted by NK cells [79].

#### 2.2.1. NK Cell Biomimetic Strategy

Toxic components in immune cells can directly induce tumor cell apoptosis, which has derived many biomimetic designs. However, their cellular uptake and intracellular transport are challenging [80]. Some strategies focus on the application of nanotechnology, such as polymersomes, to protect the toxic components (e.g., perforin and granzyme B in NK and T cells) from degradation and achieve targeted delivery [22,23,24]. Compared with pure materials and free drugs, it can induce a higher apoptotic rate, mainly due to the maintenance of structure and bioactivity, more tumor accumulation and better cellular uptake. In addition, with the assistance of targeted substances and responsive materials, the selectivity and specificity of killing can also be achieved. Based on the NK/T cell biomimetic strategy, Zhao et al. developed lysosome-targeting nanoparticles (LYS-NPs) by loading perforin and granzyme B into the metal-organic framework (MOF) and coupling with a lysosome-targeting aptamer (CD63-aptamer), to reprogram the lysosome of T cells in vivo (Figure 4a) [24]. LYS-NPs stored perforin and granzyme B in the lysosomes and released therapeutic proteins when the TCRs bound to tumor cells (Figure 4b,c). Compared with LYS-NPs and ATVs (T cell that uses lysosomes to carry anticancer proteins as adoptive T cell vectors), tumor cell apoptosis of ATVs@LYS-NPs was 2.3- and 1.2-fold higher, respectively (Figure 4d,e).

#### 2.2.2. NK Cell Engineering Strategy

Compared with T cells, NK cells recognize target cells via an array of surface ligands, without requiring prior sensitization and MHC matching [81,82]. For NK cell-based immunotherapy, the transfer safety of autologous, allogeneic, and cultured NK cells has been demonstrated [83,84,85]. However, there are still many challenges for the clinical transformation of these highly oncolytic effector cells, including in vivo life span, the poor infiltration to solid tumors [44], and the lack of inherent selectivity for the tumor. 

Materials-potentiated engineered NK cells could directly increase their cytotoxicity; for example, coating NK cells with positively charged nanoparticles to enhance their cytotoxic activity by altering the expression of CCR4 and CXCR4 of NK cells [48]. The formation of immunological synapse (IS) during cancer cell recognition plays a vital role in the NK-cell-mediated tumor killing. A reinforced NK cell (ReNK) was obtained by modifying it with acid-responsive Dox, containing a tri-block copolymer-based micellar system (Figure 5) [37]. During the binding of ReNK to cancer cells, acidification at the IS acted as a stimulus for triggering the site-specific release of Dox. As a result, ReNK could penetrate deep into the tumor tissues and kill tumor cells efficiently. 

NK cells lack an inherent targeting ability to tumor cells [40]. Besides, they are notoriously averse to endogenous gene uptake, which brings about low transgene expression for genetic engineering [86]. Current strategies mostly focus on the biological or chemical functionalization of NK cells, by using ligands [39,87], aptamers [40,41,42,44], and antibodies for increasing their migration to the tumor site and the specificity of recognizing cancer cells [43]. The specific recognition between NK cells and tumor cells brings them into proximity, thus, facilitating and prolonging intercellular interactions. The interaction between effector and target cells mediates the formation of stronger immune synapses, which facilitates the release of payloads (such as perforin, granzyme B) and the transmission of death signals. In addition, bifunctional cells have been developed to promote the migration of NK cells to target sites [87], resist the immunosuppressive microenvironment and recognize target cells [43,44]. PD-L1 expression on most tumors could be induced by IFN-γ, which results in immunosuppression. Zhang et al. developed a “self-rescue strategy” [44]. They equipped NK cells with aptamers of TLS11a and PD-L1 (T-P-NK cells) for targeting HepG2 cells and checkpoint blockade. The dual aptamer-modified T-P-NK cells showed higher affinity to HepG2 cells and secreted more IFN-γ. The tumor cell apoptosis rate rose with the increase in the effector-to-target (E/T) ratio, and significantly exceeded other groups. Notably, the percentage of apoptotic and necrotic HepG2 cells could reach up to 62.76% at the E/T ratio of 10:1 after treatment with T-P-NK cells. Additionally, the upregulated PD-L1 expression on HepG2 cells, induced by increased IFN-γ, in turn, enhanced aptamer-binding of PD-L1 for target recognition and checkpoint blockade.

Materials technology also helps to flexibly adjust these targeted molecules to obtain engineered NK cells, which have a stronger affinity with target cells. For example, modifying NK cells with different sites of the CD22 ligand can obtain analogs with micromolar affinity to CD22, thereby greatly improving the specific recognition and binding efficiency of NK cells to cancer cells [48,88]. Aptamers are a group of short and single-stranded oligonucleotides, with unique three-dimensional structures [89]. Compared with antibodies, they have many advantages, such as an inherent chemically synthetic nature and efficient tissue penetration. In addition to the native or monovalent antibody mimic (MAM), researchers synthesized a supramolecular aptamer-based polyvalent antibody mimic (PAM) on the cell surface (Figure 6a,b) [41]. In comparison to the native NK cells, cells engineered with MAMs and PAMs could form more NK-K562 complexes, thus, enhancing the efficiency of cell-cell recognition. The results showed that the enhanced interactions between the immune cells and cancer cells significantly promoted the killing of K562 cells, even at a low NK/K562 cell ratio (Figure 6c–e). Of note, the branched structure of the PAM was crucial for the enhanced cell-binding ability.

Entering the tumor site and having direct interaction with tumor cells is essential for NK cells to exhibit their cytotoxicity effect. Some studies have proven that external magnetic fields can act as a guide for NK cells. NK cells loaded with Cy5.5-coupled Fe_3_O_4_/SiO_2_ core/shell nanoparticles can increase the infiltration to B-cell lymphoma by 17 times under magnetic navigation [45]. To avoid the use of an external magnetic field, Wu et al. developed NK cells loaded with Fe_3_O_4_@PDA magnetic nanoparticles and sutured the magnetic device into the subcutaneous tissue of mice [47]. This strategy improved the accumulation and retention of NK in tumors, thereby significantly enhancing the killing effect of NK cells.

### 2.3. Neutrophil

#### 2.3.1. Neutrophil Biomimetic Strategy

For neutrophils, ROS is one of the main effect molecules for them to exhibit the cytolytic effect [90]. Many biomimetic strategies use enzymes within neutrophil biocatalytic system or enzymes with similar effects, such as superoxide dismutase (SOD), glucose oxidase (GOx), chloroperoxidase (CPO), etc., to produce highly cytotoxic ROS in tumors. However, since enzymes are easily inactivated and degraded in the physiological environment, the assistance of nanomaterials is essential [91,92]. Wu et al. used nanoscale gel to entrap the SOD and CPO [25]. With the cascade reaction of the above two enzymes, the endogenous ROS (O_2_^•−^ and H_2_O_2_) in the TME can be converted to highly cytotoxic ^1^O_2_, without external energy activation. The highly reactive HClO is the most powerful effector of neutrophils. Zhang et al. fabricated “super neutrophils” by embedding GOx and CPO into zeolitic imidazolate framework-8 (ZIF-8) and then encapsulated them with a neutrophil membrane (Figure 7a–d) [27]. In vitro and in vivo results indicated that these artificial “super neutrophils” can generate 7-fold higher reactive HClO than the natural neutrophils and reduce H_2_O_2_-involved systemic side effects via biocatalytic cascades.

In addition to the double-enzyme cascade catalytic reaction strategy that does not rely on external energy, Zhang developed an adjustable magneto-caloric-enzymatic tandem therapy [26]. Similarly, nanogels were loaded with magnetic nanoparticles (MNPs) and CPO, and the cancer cells were thermally stimulated by alternating magnetic field. The level of H_2_O_2_ or O^2•−^ could be increased, owing to the oxidative stress, followed by the upregulation of intracellular ^1^O_2_ with CPO biocatalysis. That is, the physical stimulus can be exerted as the short-term activator and the enzyme can provide the constant ROS output through biocatalysis. 

From the above review, we conclude that either internal stimulus respond to enzymatic reactions or the addition of external stimulus for precise adjustment of time and space can be utilized to integrate with the biomimetic strategy of neutrophils.

#### 2.3.2. Neutrophil Engineering Strategy

Neutrophil transfusion, the only allogeneic leukocyte transfusion, has been used in the clinical setting for decades to treat infections and used as a standard therapy [93,94,95]. However, the tumor-killing activity (TKA) of neutrophils, from more than 90% donors, is insufficient, which has greatly affected the clinical application. Therefore, it is of particular importance for neutrophils to develop methods to enhance the TKA and, thus, the immunotherapeutic effect.

In order to enhance the TKA of neutrophils, our group has developed a biotic/abiotic integration strategy, simply by integrating the nano-photosensitizer P with neutrophils (NE^P^) after incubation at 37 °C [49]. Upon NIR illumination, the improved anti-tumor effect of neutrophils was achieved by the direct synergistic effect of ROS, generated by photo-activated P (Figure 8a–f). In the biotic/abiotic integrated system, there were two sources for the effect molecule ROS: photo-activated P and intrinsic antitumor properties of neutrophils. In conventional biological approaches, the mediation of various upstream signaling pathways stimulated by cytokines is often the main reason for the enhancement of the antitumor effect of immune cells. However, we provided a materials-potentiated strategy that directly enhanced the final concentration of effect molecules, while bypassing the signaling pathways.

### 2.4. Macrophage

Macrophages account for more than 50% of tumor tissues, which are considered to be the most promising anti-tumor immune cells. However, tumor cells can escape from macrophage phagocytosis, either by expressing several anti-phagocytic markers, such as CD24 and CD47 [96,97], or by polarizing the anti-tumor M1 phenotype into the pro-tumor M2 phenotype [98]. Biomimetic and materials-potentiated macrophage engineering strategies for enhanced cancer immunotherapy are mainly designed from these two aspects. 

#### 2.4.1. Macrophage Biomimetic Strategy

Therapeutic blockade of the CD47-SIRPα pathway, to boost the antitumor activity of TAMs, has entered the clinic or is in preclinical development [99,100]. However, the safety and objective response rate need further improvement. Rao et al. developed a competitive inhibition strategy by simulating the macrophage-cancer cell interaction (Figure 9a) [28]. They wrapped magnetic nanoparticle (MN) cores, with genetically engineered cell-membrane vesicles (gCMs), overexpressing SIRPα variants, and delivered the gCMs into tumor tissues under magnetic navigation. The overexpressing SIRPα variants on the gCM shell enhanced affinity to CD47 50,000-fold, and efficiently blocked the CD47-SIRPα signaling pathway (Figure 9b). The MN core could serve to repolarize TAMs toward the M1 phenotype, synergistically facilitating the macrophage phagocytosis of cancer cells and triggering antitumor T-cell immunity.

#### 2.4.2. Macrophage Engineering Strategy

Like CAR-T, many attempts have been made to use CAR-Macrophage (CAR-M) to treat cancers [101,102]. However, the major obstacle is that macrophages struggle to keep their antitumor phenotype and become accomplices to TAMs, once injected into the body [103]. The main reasons for this, affecting their phenotype, are the metabolism dysregulation of TME, tumor debris and other tumor-infiltrating immune cells [104].

Immunoregulatory cytokines play a vital role in recovering and sustaining the anti-tumor phenotype of macrophages [105]. However, providing sufficient concentration at the target site, while minimizing side effects, remains a challenge, and free cytokines are pleiotropic, which may have adverse effects on immunotherapy. Like the T cell backpack strategy, a class of soft discoidal IFN-γ backpacks was developed [50]. The “backpacks” can firmly adhere to the surface of macrophages by the cell-adhesive layer (hyaluronic acid modified with aldehyde (HA-Ald) and poly(allylamine) hydrochloride (PAH)). The sustained release of cytokine enabled macrophages to maintain the M1 phenotype in immunosuppressive TME and polarized TAMs to the M1 phenotype, at a dose that was 100-fold lower than the maximum total dose. However, the same dose of free IFN-γ induced significantly higher IL-6 related tumor metastasis, which is also secreted by M1 macrophages [106]. Here, macrophages were not only the targets of cytokines, but also their active carriers. The inherent chemotactic ability of macrophages allows lower drug dosage to achieve the desired therapeutic effect. Simultaneously, this pre-loaded strategy provided the spatiotemporal consistency of several effects, which was conducive to the precise control of the drug dosage to maximize therapeutic outcomes.

In addition to the backpack strategy, some inorganic nanomaterials can reprogram macrophages after their phagocytosis and degradation. Superparamagnetic iron oxide nanoparticles (IONs), modified with HA, could be internalized into macrophages (HION@Macs) (Figure 10) [51]. As the IONs gradually degraded into iron ions, the production of therapeutic macrophages was stimulated. Benefiting from magnet-guided location and retention, the reprogrammed macrophages had a substantial improvement in producing bioactive components (e.g., NO and TNF-α) at the tumor site. Besides, they can endow macrophages with better resistance against M2-education compared with parent macrophages and M1 macrophages. This hybrid system combined contributions from both the biological regulation of materials and the intrinsic nature of immune cells, which was enlightening for the design of cell therapy products. Likewise, pegylated copper sulfide nanoparticles could promote the cellular ROS production of macrophages through dynamin-related protein 1 (Drp1)-mediated mitochondrial fission and direct bone marrow-derived macrophage (BMDMs) polarization towards the M1 phenotype [54]. The copper sulfide nanoparticle-stimulated BMDMs (CuS-MΦ) could maintain their low expression of CD206 for 6 days ex vivo and the intracellular Cu ions provided a key link with M1 polarization. Further, the CuS-MΦ also exhibited enhanced phagocytic and digestive ability due to the downregulated PD-1 expression. 

To make macrophages work efficiently and quickly, the injected macrophages should not only have high anti-tumor activity, but also have superior targeting properties, to reduce their circulation time and increase their interaction with tumor cells [51]. Apart from magnetic navigation to promote positioning and retention in tumors, aptamer-modified macrophages can increase the capture rate of tumor cells. Due to the improved tumor cell recognition of aptamer-modified macrophages, secretion of proinflammatory cytokines, phagocytosis of captured tumor cells, and expression of MHC class molecules were accelerated [52]. In addition, blockade of the CD47-SIRPα pathway has been widely used in promoting the efficiency of recognition and phagocytosis of cancer cells [107,108,109]. Unlike past studies that opsonize tumors with antibodies or by reprogramming TAMs, Alvey et al. engineered BMDMs with anti-SIRPa and pre-loaded human specific tumor-targeting Abs (Ab-primed Plus SIRPa Blocked macrophages, A’PB) [15]. Due to tumor-selective engorge-and-accumulate, the A’PB in tumors favored tumor regression for 1-2 weeks. Compared with the systemic injection of antibodies and donor macrophages that quickly differentiated toward non-phagocytic, high-SIRPa TAMs, these engineered macrophages had the advantages of both safety and efficiency. 

Engineered macrophages can be more sensitive to the TME and respond to it, resulting in higher efficiency and fewer side effects. In addition, the engineered macrophages could re-educate TAMs to M1 macrophages for immune recovery in most cases. Although the strategies of engineered macrophages for immunotherapy have achieved some success, there is still much work to be done, in exploring the mechanism of macrophage distribution and polarization. The discovery of these mechanisms will also provide other new ideas.

### 2.5. DC

As the most powerful antigen-presenting cells in the body, DCs can activate memory T lymphocytes and naive T lymphocytes, through effective ingestion, processing and presentation of antigens [110]. However, the immunosuppressive TME inhibits DCs’ recruitment, activation, and antigen presentation, thus, promoting tumor immune escape [111,112]. Therefore, strategies designed to improve the antigen-presenting ability of DCs are greatly needed. 

#### 2.5.1. DC Biomimetic Strategy

Due to the source and biosafety issues of clinical collection and in vitro reinfusion of DCs, an alternative strategy, artificial antigen-presenting cell (aAPC), has been proposed and rapidly developed to induce therapeutic cellular immunity. aAPC is a new technology for systemic cancer immunotherapy, based on cell engineering, or micro- or nano-particles. aAPC can mimic the natural process of biological APCs by presenting tumor-specific peptide-MHC and important signal proteins to T cells, bypassing the direct DCs activation [113]. As a synthetic system, aAPC can be maintained in an active state, without being affected by the external microenvironment and has a long shelf-life.

The activation of T cells by aAPC is affected by many factors, and the simulation of natural APC can be achieved through flexible adjustment of the following factors. First, aAPC should have three key signals as natural APCs: signal I for antigen recognition, signal II for co-stimulation and cytokines as signal III [114,115]. Second, the density, number and different combinations of these signals will directly affect the expansion and activation of T cells [116] and adequate surface contact between T cells and aAPC is essential for activation [113]. Strategies, such as shape manipulation and nanoparticle clustering, can increase the contact area with T cells and form more immune synapses [117,118]. In addition, many other factors are worthy of consideration and under constant exploration. However, the stimulation extent should be appropriately controlled to prevent the induction of exhausted T cells, which are manifested as the decreased cytokine production, reduced proliferative and killing capacity, and high expression of co-inhibitory molecules. 

In anti-tumor immunotherapy, the combination of aAPC and immune checkpoint blockade (such as anti-PD-1 [119], anti-CTLA4 [120]) has been proposed to amplify positive regulation and inhibit negative regulation. Here, the blockade of immune checkpoints is used to relieve the immune suppression of T cells, and the antigen-specific aAPC is used to effectively activate and expand T cells. The combination of the two will produce a synergistic effect, rather than a superimposed effect. 

However, manufacturing aAPCs that completely replicate the complex components and structures of DCs is almost impossible. Cell membranes that preserve some properties of natural cells, combined with synthetic materials, could make it possible to produce artificial cells [121,122,123,124]. Studies have shown that the DC membrane retains the ability to recognize cells and can achieve antigen presentation for T cell activation. Tang et al. coated nanoaggregates of the as-prepared aggregation-induced emission photosensitizers with a DC membrane (DC@AIEdots) [29]. The biomimetic nano-photosensitizers could efficiently accumulate around the tumor by hitchhiking the endogenous T cells, resulting in about a 1.6-fold increase in the tumor delivery efficiency. Upon interaction between the DC membrane ligands (MHC I, MHC II, CD80, and CD86) and T cell receptors on the surface, DC@AIEdots can induce the in vivo proliferation and activation of T cells (Figure 11). Results showed that DC@AIEdots-treated T cells induced 14-fold and 11-fold higher TNF-α or IFN-γ production, respectively, compared with the PBS buffer or bare AIEdots groups. Combining the nano-formulation that can induce tumor cell immunogenic cell death (ICD) with a DC membrane can provide tumor antigen and activate T cells, which induces an “in situ vaccine” effect. Simultaneously, the size advantage is beneficial for biomimetic DCs, migrating to the lymph nodes(LNs), where initial T cells reside. Sun et al. developed intelligent DCs (iDCs) by coating photothermal agents (IR-797) with a mature DC membrane [14]. In addition to cross-prime T cells in situ, after intratumoral injection, iDCs can also stimulate the initial T cells after migrating to tumor-draining lymph nodes. Then, these activated and expanded CD4^+^/CD8^+^ T cells secreted cytokines and reduced the expression of heat shock proteins in tumor cells, enhancing the cell damage caused by heat stress. iDCs, as a refined system, can provide a general platform for combining DCs-based immunotherapy with various materials as needed, to manufacture multifunctional nano-agents.

#### 2.5.2. DC Engineering Strategy

DCs could be engineered to enhance the antigen-presenting process for immunotherapy. Schemes have been developed to load antigens to DCs in vitro or to modify antigens with biological materials, to activate phagocytosis receptors on the surface of DCs. Although DC vaccines have been proven to be effective in many cancer types [125,126,127,128], the clinical benefits are still disappointing [129].

One of the main factors limiting their efficacy is the inefficient migration of DCs to the LNs, where DCs activate antigen-specific T cells. As previously reported, the efficiency of DCs migration, from the injection site to LNs, is usually less than 4–5% [130]. To promote DCs migration efficiency, several solutions have been attempted [131]: (1) inject pro-inflammatory factors to enhance the expression of C-C Chemokine Receptor 7 (CCR7) for attracting DCs; (2) exert stress (e.g., laser illumination, chemical stress) in the injection site to realize punctures’ enlargement in the peri-lymphatic basement membrane, accompanied by collagen fibers disarraying and cell-matrix disruption in the dermis to facilitate DCs’ migration; (3) enhance LN’s ability to recruit DCs via chemokine secretion by other immune cells, with high LNs homing ability; (4) other methods, such as changing the route of administration, including intradermal (ID) or subcutaneous (SC) routes. In addition, the introduction of materials provides new options for this dilemma. DCs loaded with magnetic nanoparticles can enhance their lymphatic targeting and subsequent anti-tumor effects, under the action of an external magnetic field [55]. Compared with the control group, magnetic pull force resulted in an 11.4- and 16-fold increase in LNs migration of BMDCs and DC2.4 cells, respectively. The low expression of tumor antigens and MHC molecules on tumors also impairs antigen uptake and presentation by DCs. Nano-DOX-loaded DCs could act as a carrier for chemotherapeutics, to increase the immunogenicity of glioblastoma (GBM) cells. The enhanced immunogenicity stimulated the maturation of DCs and lymphocytes, thereby subverting tumor-associated immunosuppression [56]. 

During DCs’ antigen presentation, the deletion of cell surface functional “arms” often leads to the failure of T cell induction, due to weakened interaction between DCs and T cells [132]. This may also be an important reason for the limited clinical effects of DCs vaccines. Therefore, in addition to focusing on regulating the interaction between DCs and antigens, effective T cell activation can also be achieved by directly improving the interaction between DCs and T cells. T cells are easy to separate from DCs, even when the DCs are mature, which is not beneficial for intercellular interactions. Using membrane-bound halo-tag protein (HTP) as an anchor, synthesized glycopolymer (poly-2-methacrylamido mannose (pMAM) or poly-2-methacrylamido glucopyranose (pMAG)), with an affinity for mannose receptors, were attached to the DCs surface [57]. The engineered DCs specifically attached to T cells through carbohydrate-lectin binding, increasing the stability of the DC-T cell complex. As a result, the frequency and duration of contact between T cells and pMAM- and pMAG-engineered DCs improved, leading to higher tumor toxicity. 

The development of biomimetic and engineered DCs can not only help to activate and expand T cells in vitro or in vivo, but also help us to further uncover the mechanism of the T cell activation better, which, in turn, helps to design better aAPC or engineered DCs.

## 3. Conclusions and Prospects

With the deep understanding of the effector mechanisms of immune cells, the biomimetic and materials-potentiated cell engineering is refreshing its meaning. The combination of cell or cell-derived components with materials combines the sensitivity and specificity of naturally occurring interactions and the maneuverability of materials, to achieve specific therapeutic effects. These strategies for anti-tumor immunotherapy include the following aspects.

(1) Use or simulate cell-derived components (e.g., intracellular proteins, death ligand, or key signals) to enhance the anti-tumor effect of biomimetic cells.

(2) Blockade the signals that are inhibiting the recognition (e.g., PD-1/PD-L1 pathway), phagocytosis (e.g., CD47-SIRPα pathway) or the killing of tumor cells (e.g., adenosine and IDO) of immune cells.

(3) Engineering immune cells with supportive components, such as cytokines and functional nanomaterials, to resist the immunoediting by the suppressive TME and maintain the anti-tumor phenotype.

(4) Engineering immune cells with tumor-targeting molecules (e.g., ligands, aptamers and antibodies) to increase the cytotoxicity efficiency.

(5) Integrating immune cells with drugs, such as chemotherapy drugs and photosensitizers to enhance their tumor killing activity.

Despite the rapid progress, there are still some challenges for biomimetic strategies to simulate immune functions of immune cells. Although the cell analogs obtained by biomimetic materials have rich functions and strong maneuverability, they are nonliving and do not have the ability to proliferate. Therefore, a large dose is often required to obtain the therapeutic concentration. Thus, most of these biomimetic strategies to simulate immune functions of immune cells are now little more than mere ideas. To promote the clinical transformation, greater efforts need to be made. Taking Krinsky and coworkers’ work as an example, they developed artificial lipid-based vesicles to synthesize anti-cancer proteins inside tumors, through containing the molecular machinery necessary for transcription and translation. This “artificial cell factory” strategy could be used as synthetic biology platforms, to synthesize therapeutic proteins on demand [133]. In order to endow the biomimetic systems with multiple functions similar to immune cells, their construction is relatively complicated. Moreover, for biomimetic strategies, the combination rules for different materials and components are not clear. Therefore, a lot of screening and simplification work needs to be done for the clinical translation. Although the research and development (R&D) road is long, once the R&D is effective, it will open new avenues for cancer treatment.

Compared with biomimetic strategies, materials-potentiated cell engineering strategies are more promising, based on the existing strategy of adoptive cell transfer technology. A clinical trial of aAPC has been conducted, using MART1/Melan-A peptide pulsed aAPC to generate CTL for treating melanoma (skin) [NCT00512889]. Despite being in the preclinical research stage, some strategies, such as the cytokine “backpack strategy” to maintain the phenotype, function and lifespan, as well as the magnetic guidance strategy to improve the tumor homing ability of immune cells, are promising [33,50,55]. For materials-potentiated cell engineering strategies, better anti-tumor efficiency is mainly determined by two aspects: the spatiotemporal consistency of the effects of functional materials and the maintaining of the vitality and function of immune cells. Thus, there are some issues that require special attention: (1) the biocompatibility of materials; (2) appropriate engineering methods; (3) controlled and tunable drug release. 

To achieve a better anti-tumor effect and promote the clinical transformation, biomimetic and materials-potentiated cell engineering strategies will continue to evolve: (1) exploration of more effective adjuvant drugs, intervention targets, and pathways related to immune suppression and activation; (2) synergistic regulation of the anti-tumor effect with multiple pathways; (3) utilizing endogenous stimulus-responded release strategies (e.g., hypoxia, acidity, various proteins); (4) thorough study of the long-term toxicity and drug release kinetics; (5) development of in vivo engineering strategies. In conclusion, the biological mechanism and function of immune cells are worthy of study and utilization in the development of cancer immunotherapy. Cell-based engineering strategies can utilize immune cells to function as a drug-carrying “ship”, to enhance their anti-tumor effects. Through smart design, not only the engineered immune cells, but also the whole immune system could be stimulated to exert the anti-tumor effect. Besides, increasing the possibility of clinical transformation and opening up new paths for cancer treatment is our ultimate goal.

## Data Availability

Not applicable.

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
