# Peer review of "Biomimetic and Materials-Potentiated Cell Engineering for Cancer Immunotherapy"

_pharmaceutics, 2022, doi:10.3390/pharmaceutics14040734_

Round 1

Reviewer 1 Report

In this review, the author summarized the research progress of nanotechnology-based biomimetic nanoparticles to improve the efficiency of immunotherapy. To add depth to the article, authors should address the following issues:

1: In the introduction, the authors should summarize the advantages of nanomaterials and why the use of nanomaterials is necessary.

2: In some places, the author's statement is inaccurate. For example, in citation 17, the original author used the platelet membrane, but the author classified it as a simulated T cell. The author should reconfirm it.

3: In DC engineering strategy, the author points out that the main challenge for efficacy is the inefficient migration of DCs to 590 lymph nodes. For this dilemma, the author should list some solutions.

4: Platelet membrane-mimicking nanoparticles also play an important role in tumor immunotherapy. The authors should also add platelet membrane-mimicking nanoparticles into this manuscript.

Reviewer 2 Report

This manuscript entitled “Biomimetic and materials-potentiated cell engineering for cancer immunotherapy” deal with many materials-engineered strategies are proposed to enhance the anti-tumor effect of immune cells in different cell types.

Authors are well suited to explain their findings by analyzing many existing literatures. Overall, the manuscript contains originality, so it is recommended that this work can be published in ' pharmaceutics'.

Author Response

Thank you for your positive comments on this manuscript.

Reviewer 3 Report

In this review the authors provide a wide overview on a rapidly developing and complex field such as that of cell engineering for cancer immunotherapy.

The review is very clear, well written and properly organized.

I have to raise only one point: reading the review I got a bit confused in the attempt to understand which methods may by more promising, or even nearly ready-to-use, and which, in contrast, are only little more than mere ideas. I would appreciate if the authors would add a paragraph, at the end of the review, trying to address this question. Also the use of a Table may be useful. This addition should render this very comprehensive review even more informative.

Reviewer 4 Report

Well written!  No significant changes.

Author Response

(The authors gave the same response as above.)
